# Clinical characteristics of patients with confirmed and asymptomatic SARS-CoV-2 infection in China

**Zongren Li[1,2][☯], Qin Zhong[2][☯], Wenyuan Li[3][☯], Dawei Zhang[4], Wenjun Wang[5], Feifei Yang[6], Kunlun He[1,2,5]***

**1** Medical Artificial Intelligence Research Center, Chinese PLA General Hospital, Beijing, P.R. China, **2** Medical Big Data Research Center, Chinese PLA General Hospital, Beijing, P.R. China, **3** School of Public Health, Zhejiang University, Hangzhou, Zhejiang, P.R. China, **4** The Fifth Medical Center, Chinese PLA General Hospital, Beijing, P.R. China, **5** Bio-engineering Research Center, Chinese PLA General Hospital, Beijing, P.R. China, **6** Department of Cardiology, Fourth Medical Center of Chinese PLA General Hospital, Beijing, P.R. China

☯ These authors contributed equally to this work.
\* kunlunhe@plagh.org

**Data Availability Statement:** All relevant data are within the paper and its Supporting Information files. Besides, we built a user-friendly webserver called "COVID Progression" (http://

## Abstract

### Objective

To examine the clinical characteristics of patients with asymptomatic novel coronavirus disease 2019 (COVID-19) and compare them with those of patients with mild disease.

### Design

A retrospective cohort study.

### Setting

Multiple medical centers in Wuhan, Hubei, China.

### Participants

A total of 3,263 patients with laboratory-confirmed severe acute respiratory syndrome coronavirus 2 (SARS-Cov-2) infection between February 4, 2020, and April 15, 2020.

### Main outcome measures

Patient demographic characteristics, medical history, vital signs, and laboratory and chest computed tomography (CT) findings.

### Results

A total of 3,173 and 90 patients with mild and moderate, and asymptomatic COVID-19, respectively, were included. A total of 575 (18.2%) symptomatic patients and 4 (4.4%) asymptomatic patients developed the severe illness. All asymptomatic patients recovered; no deaths were observed in this group. The median duration of viral shedding in asymptomatic patients was 17 (interquartile range, 9.25–25) days. Patients with higher levels of

covidprogression.ai/). The web server accepts patients' health condition indexes and provides a visualized prediction for progression risk, LoS, cost and treatment suggestions based on similarity search against the patients' medical record databases.

**Funding:** We declare no competing financial interests and this work was supported by the technology project 'Study on Comprehensive Therapy for pneumonia caused by SARS-CoV-2' [BWS2J006]. The funders had no role in study design, data collection and analysis, decision to publish, or preparation of the manuscript.

**Competing interests:** The authors have declared that no competing interests exist.

ultrasensitive C-reactive protein (odds ratio [OR] = 1.025, 95% confidence interval [CI], 1.01–1.04), lower red blood cell volume distribution width (OR = 0.68, 95% CI 0.51–0.88), lower creatine kinase Isoenzyme(0.94, 0.89–0.98) levels, or lower lesion ratio (OR = 0.01, 95% CI 0.00–0.33) at admission were more likely than their counterparts to have asymptomatic disease.

## Conclusions

Patients with younger ages and fewer comorbidities are more likely to be asymptomatic. Asymptomatic patients had similar laboratory characteristics and longer virus shedding time than symptomatic patients; screen and isolation during their infection were helpful to reduce the risk of SARS-CoV-2 transmission.

## Introduction

The pandemic of novel coronavirus disease 2019 (COVID-19) has affected over 505 million people and remains a global public health threat [1]. During the early stage of the COVID-19 outbreak, the Chinese Center for Disease Control and Prevention and other local health authorities implemented population-wide public health interventions such as close contact tracing, mass testing of close contacts, quarantine, and epidemiological investigations, regardless of the presence of symptoms or disease severity. People with mild and asymptomatic COVID-19 infections were put in medical isolation at designated cabin hospitals, allowing to observe the characteristics of asymptomatic disease progression.

Although SARS-CoV-2 infection usually presents with a fever or other respiratory symptoms, some patients with positive test findings remain symptom-free [2, 3]. Identifying asymptomatic individuals is challenging, as they tend not to seek medical assistance. An increasing body of evidence has shown no difference in viral load and potential infectivity between people with and without symptoms [4]. Some transmission models have suggested that individuals with undocumented infections accompanied with mild, limited, or no symptoms may contribute to the rapid spread of the SARS-CoV-2 virus [5, 6], suggesting a significant contribution of asymptomatic cases to disease spread. The epidemic has been contained in many countries, reopening public places. As public life gradually returns on track, asymptomatic infections should be considered a non-negligible source of infection, as they play an important role in transmission within the community. To strengthen the management of asymptomatic infected persons, it is urgent to understand the clinical characteristics of asymptomatic patients, such as duration of viral shedding and characteristics of laboratory examination, which can help identify these patients and control the spread of SARS-COV-2 among the population, which is of great significance for precise control and rapid treatment.

Because individuals infected with SARS-COV-2 may be misclassified as asymptomatic when symptoms are mild or atypical, the clinical characteristics of asymptomatic patients are unclear [7, 8]. Previous studies have focused on patients with mild symptoms or asymptomatic infections during the incubation period [9, 10]. To our knowledge, a few previous studies have examined patients who remain clinically asymptomatic after an incubation period of 14 days despite confirmed SARS-CoV-2 infection, and the timing of virus shedding in asymptomatic patients is unknown. Moreover, the characteristics of these patients have not been previously compared to those of patients that developed mild to moderate symptoms. In this study, we

aimed to examine patients that remained asymptomatic over the course of infection, including characteristics such as chest computed tomography (CT) findings and lesion ratios, to fill in the knowledge gap in the asymptomatic patients and provide evidence to inform public health policies during the post-pandemic reopening and return to work.

## Materials and methods

The present study included all eligible patients with confirmed SARS-Cov-2 infection admitted to the Huoshenshan Hospital (HSSH) and Guanggu Hospital (GGH), two hospitals that treated COVID-19 patients at the beginning of the epidemic, in Wuhan city, China, between February 4, 2020, and April 15, 2020. Patients with repeat hospital admissions during the study period and those transferred to other medical institutions were excluded. We also excluded patients with a severe or critical illness at admission and patients with laboratory data missing. Patients were then classified into symptomatic and asymptomatic groups, and the flow diagram of participant eligibility was present in Fig 1.

The Research Ethics Committee approved the study of PLA General Hospital (S2020–162–01). The Ethics Commission of the designated hospital waived the informed consent requirement for patients with emerging infectious diseases. Neither the patients nor the public was involved in the present study's design, conduct, reporting, or dissemination.

### Definition

Patients were confirmed SARS-Cov-2 infection as follows: (1) real-time fluorescent reverse transcription-polymerase chain reaction findings positive for the target nucleic acid; or (2) SARS-Cov-2-specific IgM or IgG detected in the serum. Patients were classified into symptomatic and asymptomatic groups. Asymptomatic patients were defined as individuals with a positive nucleic acid test result but without any appreciable clinical symptoms preceding 14 days and during hospitalization. Appreciable symptoms included fever, chills, respiratory symptoms (cough and shortness of breath, among others), gastrointestinal symptoms (vomiting and diarrhea, among others), and discomfort symptoms such as fatigue, anorexia, and loss of

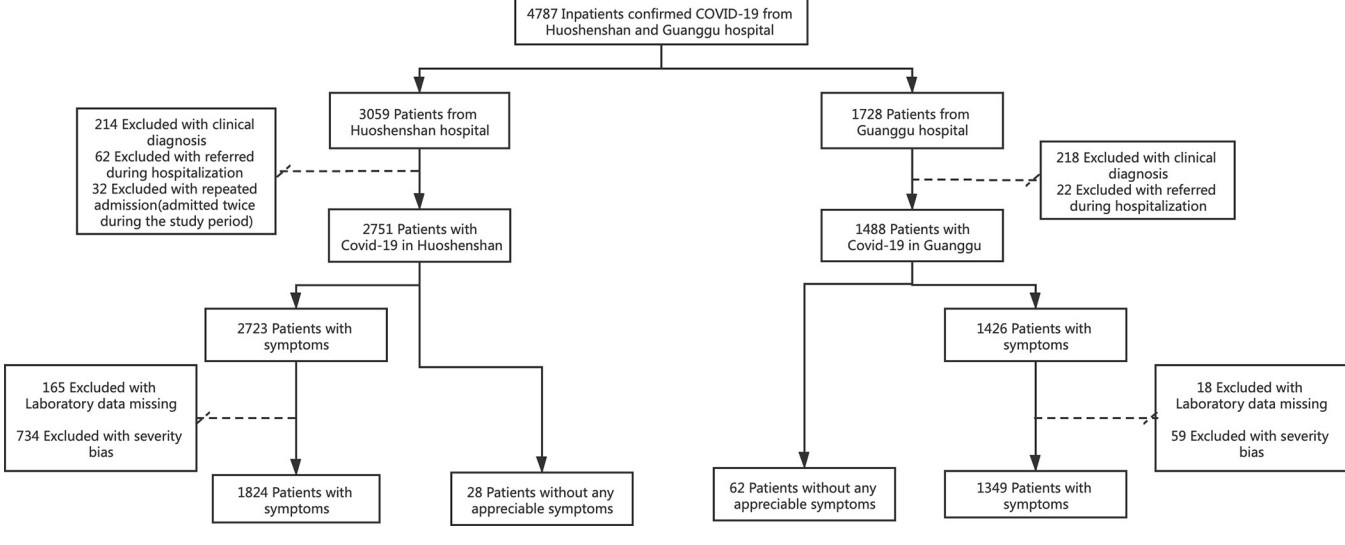

**Fig 1. Flow diagram of participant eligibility.**

taste and smell. Asymptomatic patients were detected during community investigations or medical examinations of confirmed cases' close contacts. COVID-19 severity was classified as mild, moderate, severe, and critical, according to the seventh version of the Chinese Guideline for the Management of COVID-19 published by the National Health Commission (Version six) [11]. 'Mild' disease was confirmed in the absence of imaging evidence of pneumonia and any features associated with moderate or severe disease. 'Moderate' disease was confirmed based on the evidence of pneumonia on imaging findings in the absence of other features associated with severe disease. 'Severe' disease criteria were as follows: (1) respiratory distress ($\geq$30 breaths/min); (2) oxygen saturation of $\leq$93% at rest in room air; (3) arterial partial pressure of oxygen or fraction of inspired oxygen of $\leq$ 300 mmHg (l mmHg = 0.133 kPa). Patients with 'critical' disease required mechanical ventilation and presented with septic shock or required admission to the intensive care unit. Patients were discharged only they met all the following criteria: (1) body temperature returned to normal (< 37.5˚C) for three consecutive days; (2) respiratory symptoms improved substantially; (3) pulmonary imaging showed an evident absorption of inflammation; and (4) two consecutive negative nuclei acid tests, each at least 24 h apart.

## Data collection

Data were collected from the hospitals' electronic medical records and included demographic characteristics, exposure history, symptoms, comorbidities, Charlson Comorbidity Index (CCI), baseline laboratory and CT findings, inpatient treatment log data, and length of stay.

Exposure history was defined as exposure to individuals with confirmed SARS-CoV-2 infection. Comorbidities were defined based on the International Classification of Diseases (ICD), 10th edition, clinical modification codes. The CCI was estimated based on the ICD codes [112]. All laboratory tests were performed at the study sites, following standard protocols. Baseline laboratory findings were those obtained within three days of admission. Routine blood examinations included complete blood count, coagulation profile, serum biochemical tests (including renal and liver function, creatine kinase, lactate dehydrogenase, and electrolytes), myocardial enzymes, and interleukin-6 and procalcitonin levels.

## Chest computed tomography scans and lesion ratio

All inpatients underwent chest CT scans. The treating physician determined the frequency of examinations (once a week for mild cases, once every 3–5 days for moderate cases, and once every 1–2 days for severe cases). CT examinations were performed using 64-slice CT angiography. Standard lung algorithm settings were used, specifically: mediastinal window (window width, 350 HU; window level, 40 HU); lung window (window width, 1350 HU; window level, -350 HU); and slice thickness (0.625 mm), and interlayer spacing (0.625 mm).

Under the guidance of five board-certified thoracic radiologists with >10 years of experience, an image segmentation network based on deep learning technology was used to mark the lesions on the CT images (S1 Fig). Briefly, we combined the squeeze and exception module with the residual convolution module to precisely segment ground glass, consolidation, paving stone, streak and pleural effusion findings on lung CT images. We quantified the volume of lesions on the CT scans based on segmentation results. Subsequently, we counted the number of voxels within the lung region; lung volume was obtained by multiplying the number of voxels by voxel dimensions. Voxel dimensions in the X and Y planes were the field of view divided by the image matrix size (512 × 512). The voxel dimension in the Z dimension was the slice thickness. Moreover, the observers traced the contour of the lesions on each CT slice, allowing to calculate the total lesion volume and lesion ratio (S2 Fig).

## Statistical analysis

We presented continuous variables as medians (interquartile range [IQR]) and compared the characteristics of asymptomatic and symptomatic patients using the Mann-Whitney U test. Categorical variables were presented as counts (%) and compared between groups with the chi-square test or Fisher exact test. To identify risk factors associated with being asymptomatic, univariable and multivariable logistic regression models were used. We performed multivariate imputation by chained equations if the percentage of missing values was of <30%. Given that the total number of asymptomatic patients in this study was 90, to prevent model overfitting, we applied LASSO variable selection with the optimal penalty factor λ = 0.0038, based on a 10-fold cross-validation (S3 Fig). After variable selection, the CCI score, hypertension, mean corpuscular hemoglobin concentration, red blood cell volume distribution width, lymphocyte counnd albumin, γ-glutamyl transpeptidase, lactate dehydrogenase, creatine kinase-MB, ultrasensitive C-reactive protein levels, and lesion ratio values were included in the multivariable logistic regression model. P-values of <0.05 were considered statistically significant. All statistical analyses were conducted using R software (version 4.0.3 R Foundation for statistical computing, Vienna, Austria).

## Results

A total of 3,263 COVID-19 patients with mild and moderate disease were included. There were 90 (56.7% female) asymptomatic patients and 3,173 symptomatic patients in our study. All included patients completed their in-hospital treatment and were discharged before April 15, 2020. A total of 575 (18.2%) symptomatic patients developed the severe illness. Four (4.4%) asymptomatic patients developed severe illness; no fatalities were recorded. The median age of the asymptomatic group was 55 (IQR, 39–67) years (Table 1). The symptomatic group was older, with the median age of 58 (IQR, 48–67) years. The route of transmission for all asymptomatic patients was either close contact with family members with COVID-19 or a history of exposure to epidemic areas.

In addition, in the asymptomatic group, 14 (15.6%) and 10 (11.1%) patients had hypertension and diabetes. Symptomatic patients were less likely than their asymptomatic counterparts to have hypertension (919, 29.0%, P = 0.006) and had a lower CCI score (p<0.001).

A total of 4 (4.4%) and 578 (18.2%) asymptomatic and symptomatic patients developed severe illness during hospitalization, respectively (P<0.001). The mean length of stay among asymptomatic patients was 17 (IQR 9.25–25) days.

During hospitalization, 51 (56.7%), 22 (22.2%), and 43 (47.78%) asymptomatic patients received antivirals, antibiotics, and traditional Chinese medicine products, respectively. No significant between-group differences were found among the distribution treatment types. Compared with the normal range, Asymptomatic patients had higher levels of ultrasensitive C-reactive protein (31.82%), γ-glutamyl transpeptidase (20.24%), d-dimer (10.00%), and lower counts of lymphocyte (11.11%) and neutrophil (10.00%), and increased level of procalcitonin (11.11%) (Table 2). Meanwhile, the distribution peak value of procalcitonin levels in asymptomatic patients was closer to the upper boundary of the normal range, while those of neutrophil count, total bilirubin, and creatine kinase levels were closer to the lower boundary of the corresponding normal range (Fig 2).

There were between-group differences in the levels of alanine aminotransferase (P = 0.049), aspartate aminotransferase (P = 0.022), d-dimer (P<0.001), and ultrasensitive C-reactive protein (P = 0.002), as well as in lymphocyte count (P = 0.011), albumin and creatine kinase levels, lactate dehydrogenase (P = 0.027) and γ-glutamyl transpeptidase (P = 0.004) levels, and lesion ratio (P<0.001).

**Table 1. Characteristics of 3263 patients admitted to the Huoshenshan Hospital and Guanggu Hospital between February 4, 2020, and April 15, 2020.**

| | | Symptomatic | Asymptomatic | P-value |
|---|---|---|---|---|
| | | (N = 3173) | (N = 90) | |
| Age, years | | 58 (48,67) | 55 (39,67) | 0.039 |
| Sex | | | | 0.799 |
| | Female | 1755 (55%) | 51 (57%) | |
| Length of stay, days | | 11 (7,16) | 11 (6.75, 15.25) | 0.543 |
| Duration of viral shedding | | 17(9.25,25) | 14(9.75,21.25) | 0.098 |
| Comorbidity | | | | |
| | Diabetes | 415 (13%) | 10 (11%) | 0.584 |
| | Hypertension | 919 (29%) | 14 (16%) | 0.006 |
| | Hepatitis | 72 (2%) | 1 (1%) | 0.464 |
| | Cerebrovascular | 94 (3%) | 1 (1%) | 0.303 |
| | Cancer | 46 (1%) | 3 (3%) | 0.147 |
| | CAD | 172 (5%) | 4 (4%) | 0.686 |
| | COPD | 20 (1%) | 0 (0%) | 0.450 |
| Charlson Comorbidity Index score | | | | <0.001 |
| | 0 | 2118 (67%) | 72 (80%) | |
| | 1 | 674 (21%) | 16 (18%) | |
| | >1 | 381 (12%) | 2 (2%) | |
| Treatments | | | | |
| Antiviral therapy | | 1797 (57%) | 51 (57%) | 0.095 |
| Antibiotic drug | | 1006 (32%) | 20 (22%) | 0.056 |
| Immune globulin | | 403 (13%) | 11 (12%) | 0.893 |
| Glucocorticoid therapy | | 240 (8%) | 4 (4%) | 0.267 |
| Traditional Chinese medicine | | 2143 (68%) | 43 (48%) | <0.001 |
| Develop Severe illness | | | | <0.001 |
| | Yes | 578 (18%) | 4 (4%) | |

Data are presented as median (IQR), n (%). P-values were derived from the Mann-Whitney U test, $\chi^2$ test, or Fisher exact test, as suitable. CAD = coronary heart disease; COPD = chronic obstructive pulmonary disease. Duration of viral shedding is defined as the duration from the first to last positive nasopharyngeal swab.

Upon admission, CT scans revealed focal ground-glass opacities in 52 asymptomatic individuals (52/88, 59.1%) and stripe shadows and/or diffuse consolidation in 27 individuals (27/88, 30.7%), whereas 20 individuals (20/88, 22.7%) had no abnormalities. Two patients' CT scans were not recorded. Abnormal radiological findings confined to one or both lungs were identified in 75% (51/68) and 25% (17/68) of the asymptomatic individuals, respectively.

In univariable analysis (Table 3), the odds of asymptomatic infections were lower in patients that were young (odds ratio [OR] 0.98, 95% confidence interval [CI] 0.97–0.99) and those that had hypertension (OR = 0.45, 95% CI 0.24–0.78). Mean corpuscular hemoglobin concentration values (OR = 0.62, 95% CI 0.47–0.79), and albumin (OR = 1.11, 95% CI 1.05–1.18), lactate dehydrogenase (OR = 0.99, 95% CI 0.99–1.00), and creatine kinase-MB (OR = 0.92, 95% CI 0.87–0.97) levels, and lesion ratio were associated with being asymptomatic(OR = 0.00, 95% CI 0.00–0.01). Multivariable logistic regression analysis results are presented in Table 3. The receiver operating characteristic curve of the multivariable logistic regression model is shown in S4 Fig. We found that higher ultrasensitive C-reactive protein levels (OR = 1.03, 95%, CI 1.01–1.04), lower red blood cell volume distribution width (OR = 0.68, 95% CI 0.51–0.88), creatine kinase-MB (OR = 0.94, 95% CI 0.89–0.98) values, and

**Table 2. Basic laboratory indicators of the study population.**

| | Symptomatic | Asymptomatic | Normal Range | P-value |
|---|---|---|---|---|
| | (N = 3173) | (N = 90) | | |
| Procalcitonin (ng/ml) | 0.04 (0.04, 0.06) | 0.04 (0.04, 0.05) | 0–0.05 | 0.750 |
| | H: 523 (26%) | H: 10 (11%) | | |
| White blood cell count ($*10^9$/L) | 5.6 (4.7,6.8) | 6 (5.1, 7.0) | 3.5–9.5 | 0.105 |
| | L: 138 (4%) | L: 1 (1%) | | |
| | H: 124 (4%) | H: 0 (0%) | | |
| Mean corpuscular hemoglobin concentration (g/L) | 338 (332, 343) | 341 (335, 347) | 320–360 | <0.001 |
| | L: 104 (3%) | L: 2 (2%) | | |
| | H: 18 (1%) | H: 0 (0%) | | |
| Red blood cell volume distribution width (%) | 12.9 (12.4, 13.5) | 12.5 (12.1, 13.0) | 11.0–16.0 | <0.001 |
| | L: 3 (0%) | L: 0 (0%) | | |
| | H: 90 (3%) | H: 1 (1%) | | |
| Platelet count ($*10^9$/L) | 221 (183, 266) | 217 (183, 253) | 125–320 | 0.769 |
| | L: 142 (5%) | L: 1 (1%) | | |
| | H: 338 (11%) | H: 4 (4%) | | |
| Basophil percentage (%) | 0.4 (0.3,0.5) | 0.4 (0.3, 0.5) | 0.0–1.0 | 0.596 |
| | H: 80 (3%) | H: 2 (2%) | | |
| Neutrophil count ($*10^9$/L) | 3.4 (2.7, 4.3) | 3.5 (2.9, 4.3) | 2.5–7.5 | 0.303 |
| | L: 623 (20%) | L: 9 (10%) | | |
| | H: 89 (3%) | H: 0 (0%) | | |
| Lymphocyte count ($*10$-9/L) | 1.6 (1.3, 1.9) | 1.7 (1.3, 2.2) | 1.1–3.2 | 0.005 |
| | L: 543 (17%) | L: 10 (11%) | | |
| | H: 35 (1%) | H: 2 (2%) | | |
| Basophil count ($*10$-9/L) | 0.02 (0.01, 0.03) | 0.02 (0.02, 0.03) | 0–0.06 | 0.624 |
| | H: 61 (2%) | H: 0 (0%) | | |
| Alanine aminotransferase (IU/L) | 21.2 (13.9, 34.6) | 18.5 (11.4, 27.8) | 0.0–40.0 | 0.049 |
| | H: 583 (20%) | H: 13 (15%) | | |
| Aspartate aminotransferase (IU/L) | 17.5 (13.8, 23.6) | 16 (12.7, 20.4) | 0.0–45.0 | 0.022 |
| | H: 133 (4%) | H: 4 (5%) | | |
| Albumin (g/L) | 38.9 (36.4, 41.1) | 40.3 (38.7, 42.3) | 35.0–51.0 | <0.001 |
| | L: 445 (15%) | L: 6 (7%) | | |
| | H: 4 (0%) | H: 0 (0%) | | |
| Albumin/globulin ratio (%) | 1.3 (1.2, 1.5) | 1.37 (1.2, 1.5) | 1–2.4 | 0.082 |
| | L: 174 (7%) | L: 7 (9%) | | |
| Total bilirubin (umol/L) | 9.4 (7.2, 12.3) | 9.7 (7.2, 13.2) | 1.7–21.0 | 0.385 |
| | H: 119 (4%) | H: 5 (6%) | | |
| Creatinine (umol/L) | 63 (55, 75) | 63 (54, 72) | 44–106 | 0.478 |
| | L: 94 (3%) | L: 2 (2%) | | |
| | H: 74 (3%) | H: 0 (0%) | | |
| Alkaline phosphatase (IU/L) | 69 (58, 83) | 68 (57, 84) | 40–130 | 0.537 |
| | L: 69 (2%) | L: 2 (2%) | | |
| | H: 71 (2%) | H: 1 (1%) | | |
| γ-glutamyl transpeptidase (IU/L) | 26.6 (18.2, 43.3) | 23.0 (16.0, 32.3) | 0.0–40.0 | 0.004 |
| | H: 832 (28%) | H: 17 (20%) | | |
| Creatine Kinase (IU/L) | 53 (39, 74) | 64 (51, 84) | 24–170 | <0.001 |
| | L: 111 (5%) | L: 0 (0%) | | |
| | H: 52 (2%) | H: 2 (4%) | | |

*(Continued)*

**Table 2.** (Continued)

| | Symptomatic | Asymptomatic | Normal Range | P-value |
|---|---|---|---|---|
| | (N = 3173) | (N = 90) | | |
| Lactate dehydrogenase (IU/L) | 171 (149, 201) | 160 (141, 185) | 120–250 | 0.027 |
| | L: 89 (4%) | L: 3 (6%) | | |
| | H: 214 (10%) | H: 1 (2%) | | |
| Creatine kinase-MB (IU/L) | 6.9 (0.7,9.4) | 1.3 (0.6, 8.5) | 0.0–24.0 | 0.019 |
| | H: 31 (1%) | H: 0 (0%) | | |
| D-dimer (mg/L) | 0.33 (0.19, 0.62) | 0.21 (0.13, 0.36) | 0.00–0.55 | <0.001 |
| | H: 649 (28%) | H: 9 (10%) | | |
| Ultrasensitive C-reactive protein (mg/L) | 1.6 (0.6, 4.4) | 1.0 (0.4, 2.4) | 0.0–4.0 | 0.002 |
| | H: 791 (27%) | H: 14 (32%) | | |
| IL-6 (pg/ml) | 1.5 (1.5, 2.3) | 1.5 (1.5, 1.5) | 0.0–7.0 | 0.112 |
| | H: 141 (10%) | H: 2 (5%) | | |
| Lesion-ratio (%) | 0.028 | 0.002 | 0.000–0.010 | <0.001 |
| | (0.004, 0.1018) | (0.00, 0.007) | | |

Data are presented as medians (IQR), n (%). P-values were derived the by Mann-Whitney U test, χ2 test, or Fisher exact test, as suitable. H: n (%) represents the number and percentage of patients with values above the normal range, L: n (%) represents the number and percentage of patients with values below the normal range.

lesion ratio (OR = 0.00, 95% CI 0.00–0.33) at admission were associated with higher odds of asymptomatic disease.

## Discussion

In this retrospective multicenter study, we aimed to report the detailed clinical characteristics of asymptomatic patients and their comparison with normal range and symptomatic patients, which could help provide a better approach to the outbreak response and assessment. We found that asymptomatic patients had longer virus shedding time and younger age than symptomatic patients, and the proportion of patients with hypertension was significantly smaller than symptomatic patients. In addition, asymptomatic patients also had lung lesions which were relatively mild compared with symptomatic patients. Moreover, although without clinical symptoms, asymptomatic patients had similar laboratory characteristics, which made it difficult for us to identify asymptomatic patients in screening.

Previous studies reported that the median duration of viral shedding in severe and critical cases was 19.0 and 24.0 days [12], respectively, while the corresponding value in patients with mild illness was 14 days, suggesting that patients with mild illness may clear the virus in a relatively short time [4, 13]. In the present study, the mean duration of virus shedding in the asymptomatic group was 17 days, which was shorter than that in the severe/critical group and longer than that in patients with mild disease in previous reports. Meanwhile, a recent study found similar viral load values in asymptomatic and symptomatic patients [4]. These findings suggest that performing COVID-19 screening only for symptomatic individuals may miss asymptomatic cases, increasing the rate of infection transmission; asymptomatic patients should be screened, identified, and isolated to reduce the risk of disease transmission. Multivariable regression revealed that patients with lower red blood cell volume distribution width, lower creatine kinase-MB values, higher ultrasensitive C-reactive protein levels, and higher lesion ratios were more likely to be asymptomatic than their counterparts, although they are all within the normal range. Compared with mild and moderate symptomatic patients admitted to the hospital, asymptomatic patients have similar laboratory characteristics although they

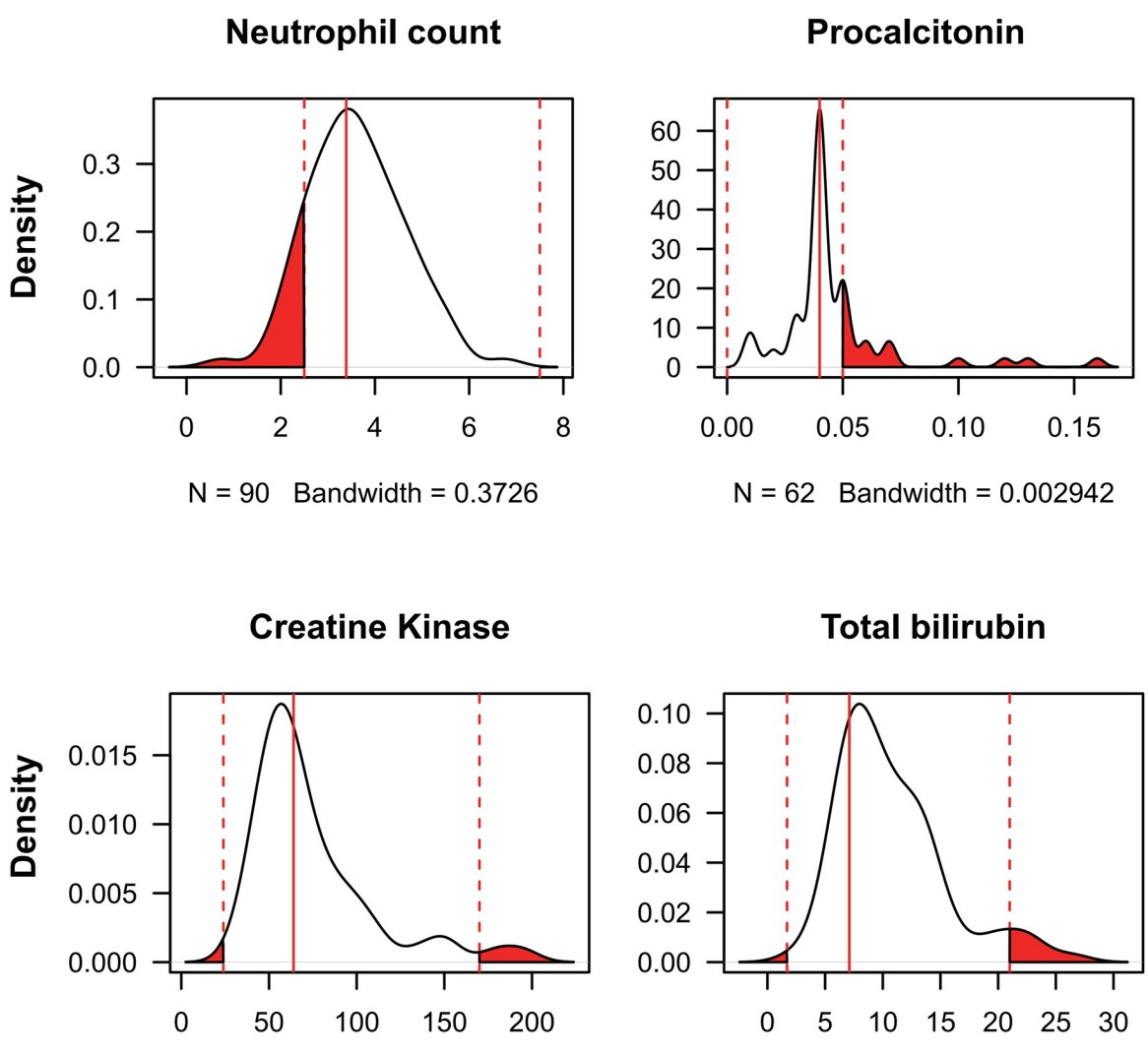

*Left and right dotted vertical red lines indicate upper and lower bounds of normal conditions for each biomedical index respectively. Solid vertical red line indicates mode of each biochemical index among the patients. Red area under the density curves indicates abnormal cases out of range of the normal condition standards for each biochemical index*

**Fig 2. Density distribution of blood biochemical index values in COVID-19 patients with asymptomatic infection.**

have no clinical symptoms, which brings us difficulties in screening asymptomatic patients. Moreover, asymptomatic patients have lung lesions; these changes may be relatively mild. Yanli et al. reported that more than half of 74 cases with asymptomatic infections presented with ground-glass opacities in the lung on chest CT scans [14]. These findings are consistent with those of the present study.

The strength of this study is that it systematically examined the clinical characteristics of patients with asymptomatic COVID-19 over the course of their hospitalization and that it compared these characteristics with those of symptomatic patients. To our knowledge, no

**Table 3. Univariable and multivariable logistic regression results for the risk of asymptomatic disease among inpatients with COVID-19.**

| | Univariable OR (95% CI) | P-value | Multivariable OR (95% CI) | P-value |
|---|---|---|---|---|
| Length of stay, days | 1.01 (0.98–1.04) | 0.462 | | |
| CCI-Score | | | | |
| 1 | 0.70 (0.39–1.18) | 0.200 | 1.14 (0.63–2.13) | 0.682 |
| >1 | 0.26 (0.04–0.84) | 0.063 | 0.38 (0.09–1.6) | 0.197 |
| Sex | 0.95 (0.62–1.44) | 0.799 | | |
| Age, years | 0.98 (0.97–0.99) | 0.006 | | |
| Comorbidity | | | | |
| CAD | 0.81 (0.25–1.97) | 0.687 | | |
| Cancer | 2.34 (0.56–6.57) | 0.160 | | |
| Cerebrovascular | 0.37 (0.02–1.68) | 0.323 | | |
| Diabetes | 0.83 (0.40–1.54) | 0.777 | | |
| Hepatitis | 0.48 (0.03–2.23) | 0.473 | | |
| Hypertension | 0.45 (0.24–0.78) | 0.007 | 0.60 (0.30–1.13) | 0.125 |
| White blood cell count (*10~9/L) | 1.02 (0.91–1.12) | 0.768 | | |
| Mean corpuscular hemoglobin concentration (g/L) | 1.02 (1.00–1.03) | 0.014 | 1.01 (0.99–1.03) | 0.295 |
| Red blood cell volume distribution width (%) | 0.62 (0.47–0.79) | <0.001 | 0.68 (0.51–0.88) | 0.005 |
| Platelet count (*10-9/L) | 1.00 (0.99–1.00) | 0.475 | | |
| Basophil percentage (%) | 0.64 (0.25–1.48) | 0.32 | | |
| Neutrophil count | 0.97 (0.84–1.09) | 0.632 | | |
| Lymphocyte count | 1.65 (1.21–2.26) | 0.002 | 1.37 (0.93–1.97) | 0.099 |
| Alanine aminotransferase (IU/L) | 1.00 (0.99–1.00) | 0.295 | | |
| Aspartate aminotransferase (IU/L) | 0.98 (0.96–1.00) | 0.097 | | |
| Albumin (g/L) | 1.11 (1.05–1.18) | <0.001 | 1.058 (0.99–1.13) | 0.116 |
| Albumin /globulin ration (%) | 1.26 (0.62–1.96) | 0.392 | | |
| Total bilirubin (umol/L) | 1.01 (0.97–1.03) | 0.682 | | |
| Creatinine (umol/L) | 0.99 (0.98–1.00) | 0.294 | | |
| Alkaline phosphatase (IU/L) | 1.00 (0.99–1.01) | 0.954 | | |
| γ-glutamyl transpeptidase (IU/L) | 0.99 (0.98–1.00) | 0.105 | 1.00 (0.99–1.00) | 0.232 |
| Lactate dehydrogenase (IU/L) | 0.99 (0.99–1.00) | 0.028 | 1.00 (0.99–1.00) | 0.589 |
| Creatine kinase-MB (IU/L) | 0.92 (0.87–0.97) | 0.003 | 0.94 (0.89–0.98) | 0.010 |
| Prothrombin time (s) | 1.00 (0.84–1.13) | 0.989 | | |
| ultrasensitive C-reactive protein (mg/L) | 1.007 (0.99–1.02) | 0.354 | 1.03 (1.01–1.04) | <0.001 |
| IL-6 (pg/ml) | 0.90 (0.73–0.98) | 0.203 | | |
| D-dimer (mg/L) | 0.65 (0.36–0.97) | 0.101 | | |
| Procalcitonin (ng/ml) | 0.01 (0.00–0.54) | 0.219 | | |
| Lesion-ratio (%) | 0.00 (0.00–0.01) | <0.001 | 0.01 (0.00–0.33) | 0.023 |

study to date has examined the differences in the duration of viral shedding and clinical characteristics between symptomatic and asymptomatic patients. In addition, the present study has reported detailed chest CT findings of the included patients. However, this study has several limitations. First, only 90 asymptomatic patients with confirmed COVID-19 were included, while the number of symptomatic patients was up to 3173. The imbalance of sample size may lead to a decrease in the accuracy of the results, and larger studies are thus required to validate the present findings. Second, during the screening process, some patients failed to report pertinent information regarding their symptoms, as they were unaware of its relevance, which may have resulted in misclassification. Future studies are required to examine the physical and pathological characteristics of COVID-19 patients that correspond to their clinical features. As

this was a retrospective study, some information regarding inflammation markers such as d-dimer and interleukin-6 levels was incomplete due to the limited resources available to isolated patients during the ongoing pandemic. As the missing data were likely missing at random, we used multiple imputations to account for them. Lastly, we did not collect data after discharge. Further studies are required to evaluate the long-term effects of COVID-19 in asymptomatic patients.

In conclusion, patients with younger ages and fewer comorbidities are more likely to be asymptomatic. Asymptomatic patients had similar laboratory characteristics and longer virus shedding time than symptomatic patients; screen and isolation during their infection were helpful to reduce the risk of SARS-CoV-2 transmission. All the results are publicly available on the website(http://covidprogression.ai/).

## Supporting information

**S1 Fig. Comparison of CT scans before and after expert annotation.**
(DOCX)

**S2 Fig. Chest CT scans from six patients with different lesion-ratio.**
(DOCX)

**S3 Fig. The least absolute shrinkage and selection operator (LASSO) binary logistic regression model.** (a) LASSO coefficient profiles of the 36 baseline features. (b) Tuning parameter (λ) selection in the LASSO model used 10-fold cross-validation via minimum criteria.
(DOCX)

**S4 Fig. ROC curve of the multivariable logistic regression model.**
(DOCX)

## Author Contributions

**Conceptualization:** Zongren Li, Wenyuan Li, Kunlun He.

**Data curation:** Qin Zhong, Wenjun Wang.

**Investigation:** Qin Zhong.

**Project administration:** Zongren Li.

**Resources:** Kunlun He.

**Validation:** Wenyuan Li, Dawei Zhang, Wenjun Wang, Feifei Yang.

**Visualization:** Wenjun Wang.

**Writing – original draft:** Zongren Li, Qin Zhong, Wenyuan Li.

**Writing – review & editing:** Wenyuan Li, Dawei Zhang, Kunlun He.

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
