## [Decision Letter · Decision Letter 0]

3 Mar 2022

PONE-D-21-19790

Clinical characteristics of patients with confirmed and asymptomatic SARS-CoV-2 infection in China

PLOS ONE

Dear Dr. He,

Thank you for submitting your manuscript to PLOS ONE. After careful consideration, we feel that it has merit but does not fully meet PLOS ONE’s publication criteria as it currently stands. Therefore, we invite you to submit a revised version of the manuscript that addresses the points raised during the review process.

Please revise.

We look forward to receiving your revised manuscript.

Kind regards,

Academic Editor

PLOS ONE

Journal Requirements:

2. Thank you for stating the following financial disclosure: "The funders had no role in study design, data collection and analysis, decision to publish, or preparation of the manuscript."

Reviewers' comments:

Reviewer's Responses to Questions

**Comments to the Author**

1. Is the manuscript technically sound, and do the data support the conclusions?

Reviewer #1: Partly

Reviewer #2: Partly

Reviewer #3: Yes

2. Has the statistical analysis been performed appropriately and rigorously? 

Reviewer #1: Yes

Reviewer #2: Yes

Reviewer #3: Yes

3. Have the authors made all data underlying the findings in their manuscript fully available?

Reviewer #1: Yes

Reviewer #2: Yes

Reviewer #3: Yes

4. Is the manuscript presented in an intelligible fashion and written in standard English?

Reviewer #1: Yes

Reviewer #2: Yes

Reviewer #3: No

5. Review Comments to the Author

Reviewer #1: The article, "Clinical characteristics of patients with confirmed and asymptomatic SARS-CoV-2

infection in China" reads well and is current given the context. The Introduction section is drafted well however the rationale for why there is a need to understand the clinical characteristics of patients with asymptomatic SARS-CoV-2 infection could be further strengthened.

The author in brief could state the reason for selection of the two health care centres/hospital, that being Huoshenshan Hospital (HSSH) and Guanggu Hospital (GGH) in Wuhan city. Were they the only hospitals treating the COVID-19 cases in Wuhan?

Also, the process of arriving at the said sample size and the sampling method could be explicitly stated in the article for reproducibility. Given the methodology, there is an introduction of certain biases within the study, one of which is misclassification bias which is already stated in the article under the limitations.

Furthermore, the sample size arrived at for asymptomatic COVID-19 cases (90) vs the symptomatic (3173) is too small to generalize the findings and draw any said conclusions especially considering that approximately 80% SARS-CoV-2 infective cases are of the asymptomatic nature.

The article could use some further editing and the references are to adhere to the Vancouver style formatting.

Reviewer #2: The authors have studied the clinical characteristics of patients with asymptomatic novel coronavirus disease 2019 (COVID-19) and compare them with those of patients with mild disease in a retrospective cohort. They found that the risk of asymptomatic infection was higher among patients with lower red blood cell volume distribution width, creatine kinase-MB levels, and lesion ratio, and higher hypersensitive C-reactive protein levels than among their counterparts. Heightened surveillance using laboratory and chest CT findings among these individuals may improve the detection of unrecognized SARS-CoV-2 infection. The work is good and could clarify the pandemic. Below are a few comments.

Comments:

1. Kindly describe the inclusion and exclusion criteria of the patients and the controls in the study in detail.

2. Line 182-190, the level of neutrophil etc. in asymptomatic patient were higher compared to which group?

3. Table 2, most of the values are comparable between symptomatic and asymptomatic patients and those were under normal range, so what we could say about it?

4. The significance of the study is not clear. Kindly add suitable data to draw some conclusion.

5. Discussion could be improved by enrichment of available literature and comparing the findings with the contemporary works. What is the take home message of the study?

Reviewer #3: The investigators analysed the data from symptomatic and asymptomatic cases of SARS-CoV-2 infection in Wuhan, China. As the study data brought out several important findings to the medical literature the manuscript may be accepted with minor revision as follows:

1. The conclusive statement in the abstract and at the end of manuscript may be modified to state that “asymptomatic infection was associated with …… “ rather than to state the risk of asymptomatic …..

2. “Chest CT findings among these individuals may improve the detection of unrecognized SARS-CoV-2 infection” – this statement may be inappropriate as chest CT findings may not be the best choice or indicative for detecting asymptomatic cases.

3. In Tables 1 and 3, including the data of common combination of risk factors such as Diabetes and Hypertension might give additional information.

4. “SARS-Cov-2-specific IgM or IgG detected in the serum” stated in Materials and methods, but no data is found in results.

5. Instead of using Hypersensitive, may be Ultrasensitive CRP might be the correct and acceptable terminology.

6. PLOS authors have the option to publish the peer review history of their article (what does this mean?). If published, this will include your full peer review and any attached files.

Reviewer #1: No

Reviewer #2: **Yes: **Mohammad Asad

Reviewer #3: No

---

## [Author Response · Author response to Decision Letter 0]

1 Jun 2022

Re: PONE-D-21-19790

Dear Editor,

We are submitting our revised manuscript entitled “Clinical characteristics of patients with confirmed and asymptomatic SARS-CoV-2 infection in China” for your consideration. 

We are grateful for the constructive feedback provided by peer reviewers. Please see our responses to specific comments below. Line numbers in our responses correspond to those in the tracked changes version of our revised manuscript.

We hope these revisions are to your satisfaction. Please let us know if you have any further questions or comments.

Sincerely,

Kunlun He, MD, PhD.

Professor, Medical Big Data Research Center, Chinese PLA General Hospital, Beijing 100039, China.

Email: kunlunhe@plagh.org.

 

Response to Reviewers

Response to Reviewer #1: 

1. Comment: The article, "Clinical characteristics of patients with confirmed and asymptomatic SARS-CoV-2 infection in China" reads well and is current given the context. The Introduction section is drafted well however the rationale for why there is a need to understand the clinical characteristics of patients with asymptomatic SARS-CoV-2 infection could be further strengthened.

Response: Thank you for your recognition of our work and advice on the introduction section, we have reorganized our introduction and strengthened the importance of reporting the clinical characteristics of asymptomatic patients (Lines 75-82). Specifically, as public life gradually returns on the track, asymptomatic infections should be considered a non-negligible source of infection, because they played an important role in transmission within the community. To strengthen the management of asymptomatic infected persons, it is urgent to understand the clinical characteristics of asymptomatic patients, such as duration of viral shedding and characteristics of laboratory examination, which can help identify these patients and control the spread of SARS-COV-2 among the population, which is of great significance for precise control and rapid treatment.

2. Comment: The author in brief could state the reason for selection of the two health care centres/hospital, that being Huoshenshan Hospital (HSSH) and Guanggu Hospital (GGH) in Wuhan city. Were they the only hospitals treating the COVID-19 cases in Wuhan?

Response: In the method part, we briefly explained the reasons why we chose these two hospitals (Lines 110-158). specifically, these two hospitals were hospitals with more patients in Wuhan at the beginning of the epidemic outbreak. at the same time, HSSH is an emergency infectious disease hospital (usually treating patients with complex conditions), and GGH is a square cabin hospital (usually treating mild and asymptomatic patients). These two hospitals cover different patients and were representative to some extent.

3. Comment: the process of arriving at the said sample size and the sampling method could be explicitly stated in the article for reproducibility. 

Response: As shown in figure 1, we use a full sampling method when selecting patients, which means we include all patients who meet the requirements. We revised the method section to emphasize that.

4. Comment: Furthermore, the sample size arrived at for asymptomatic COVID-19 cases (90) vs the symptomatic (3173) is too small to generalize the findings and draw any said conclusions, especially considering that approximately 80% SARS-CoV-2 infective cases are of the asymptomatic nature.

Response: First, compared with asymptomatic patients in a broad sense (including patients with mild symptoms or in the incubation period), our study adopted the definition that patients without any appreciable clinical symptoms throughout the process, which is a relatively stringent definition. Secondly, unlike Omicron or Delta, we included the first group of patients who were infected with COVID-19. Among these patients, the number of asymptomatic patients who meet our definition is relatively small. And in other reports of the early prevalence of COVID-19[1,2], the asymptomatic rate of Chinese patients is about 1.5% to 2.8%, and our study is also in line with this proportion.

5. Comment: The article could use some further editing and the references are to adhere to the Vancouver style formatting.

Response: We have changed the references to Vancouver style according to your request.

Response to Reviewer #2: 

1. Comment: Kindly describe the inclusion and exclusion criteria of the patients and the controls in the study in detail

Response: We have reorganized the methods section on the standard of nano-platoon. Specifically, we included all eligible patients with positive nasopharyngeal swabs who admitted to the Huoshenshan Hospital (HSSH) and Guanggu Hospital (GGH). Patients with repeat hospital admissions during the study period and those transferred to other medical institutions were excluded. We also excluded patients with a severe or critical illness at admission and patients with laboratory data missing. (Lines 110-163)

2. Comment: Line 182-190, the level of neutrophils etc. in asymptomatic patient were higher compared to which group?

Response: In this part, we meant that, compared with the normal range, asymptomatic patients had higher levels the level of neutrophils etc. We also made changes in manuscript to make the statement clear (Lines 394-395).

3. Comment: In table 2, most of the values are comparable between symptomatic and asymptomatic patients and those were under normal range, so what we could say about it?

Response: In this part, we found asymptomatic patients have similar laboratory characteristics with symptomatic patients although they have no clinical symptoms. Even though multivariable regression revealed that patients with lower red blood cell volume distribution width, lower creatine kinase-MB values, higher ultrasensitive C-reactive protein levels, and higher lesion ratios were more likely to be asymptomatic than their counterparts, they are all within the normal range. Thus, we revised the conclusion and discussion parts.

4. Comment: The significance of the study is not clear. Kindly add suitable data to draw some conclusion.

Response: We reorganized our conclusion and add more details such as the duration of viral shedding to draw this (Lines 284-285; Lines 340-349).

5. Comment: Discussion could be improved by enrichment of available literature and comparing the findings with the contemporary works. What is the take home message of the study?

Response: Thank you for your suggestion, we have revised the relevant part, in short, we found that patients with younger ages and fewer comorbidities are more likely to be asymptomatic. Asymptomatic patients had similar laboratory characteristics and longer virus shedding time than symptomatic patients, screen and isolation during their infection are helpful to reduce the risk of SARS-CoV-2 transmission.

Response to Reviewer #3: 

1. Comment: The conclusive statement in the abstract and at the end of manuscript may be modified to state that “asymptomatic infection was associated with …… “rather than to state the risk of asymptomatic …..

Response: Thank you for your suggestion, we have revised our conclusions and abstracts following your comments (Lines 48-52; Lines 340-349; Lines 418-424), and the new conclusions will focus more on the comparison between asymptomatic infections and normal and symptomatic patients.

2. Comment: “Chest CT findings among these individuals may improve the detection of unrecognized SARS-CoV-2 infection” – this statement may be inappropriate as chest CT findings may not be the best choice or indicative for detecting asymptomatic cases.

Response: Thank you for your advice. We have modified the relevant parts (Lines 345-348). In short, we have found that although some patients do not have any clinical symptoms, they still have lung lesions, and the proportion of lesions is lower than that of mild patients.

3. Comment: In Tables 1 and 3, including the data of common combination of risk factors such as Diabetes and Hypertension might give additional information.

Response: Thank you for your suggestions, we used the Charlson Comorbidity Index (CCI) as an indicator to evaluate the comorbidities with patients, and found that patients with fewer comorbidities were more likely to become asymptomatic infections, and added relevant content in manuscript (Lines 48-51; 289-290; 420-423).

4. Comment: “SARS-Cov-2-specific IgM or IgG detected in the serum” stated in Materials and methods, but no data is found in results.

Response: Although we considered SARS-Cov-2-specific IgM or IgG when we include patients. In fact, only a small number (11%) of patients whose initial nucleic acid tests were negative have those data, and these patients also show positive in nucleic acid tests after admission, and the specific data can be obtained from the open-source website (http://covidprogression.ai/) we provided.

5. Comment: Instead of using Hypersensitive, may be Ultrasensitive CRP might be the correct and acceptable terminology.

Response: Thank you for your correction. We have replaced it in the full text.

REFERENCES

1. Chen Z, Wang B, Mao S, Ye Q. Assessment of global asymptomatic SARS-CoV-2 infection and management practices from China. Int J Biol Sci. 2021;17(4):1119-1124. Published 2021 Mar 10. doi:10.7150/ijbs.59374

2. Wu, Zunyou, and Jennifer M McGoogan. “Asymptomatic and Pre-Symptomatic COVID-19 in China.” Infectious diseases of poverty vol. 9,1 72. 22 Jun. 2020, doi:10.1186/s40249-020-00679-2

---

## [Decision Letter · Decision Letter 1]

27 Jun 2022

PONE-D-21-19790R1Clinical characteristics of patients with confirmed and asymptomatic SARS-CoV-2 infection in ChinaPLOS ONE

Dear Dr. He,

Thank you for submitting your manuscript to PLOS ONE. After careful consideration, we feel that it has merit but does not fully meet PLOS ONE’s publication criteria as it currently stands. Therefore, we invite you to submit a revised version of the manuscript that addresses the points raised during the review process.

Please revise.

We look forward to receiving your revised manuscript.

Kind regards,

Academic Editor

PLOS ONE

Journal Requirements:

Reviewers' comments:

Reviewer's Responses to Questions

**Comments to the Author**

1. If the authors have adequately addressed your comments raised in a previous round of review and you feel that this manuscript is now acceptable for publication, you may indicate that here to bypass the “Comments to the Author” section, enter your conflict of interest statement in the “Confidential to Editor” section, and submit your "Accept" recommendation.

Reviewer #1: All comments have been addressed

Reviewer #2: All comments have been addressed

2. Is the manuscript technically sound, and do the data support the conclusions?

Reviewer #1: Yes

Reviewer #2: Yes

3. Has the statistical analysis been performed appropriately and rigorously? 

Reviewer #1: Yes

Reviewer #2: I Don't Know

4. Have the authors made all data underlying the findings in their manuscript fully available?

Reviewer #1: Yes

Reviewer #2: Yes

5. Is the manuscript presented in an intelligible fashion and written in standard English?

Reviewer #1: Yes

Reviewer #2: Yes

6. Review Comments to the Author

Reviewer #1: Vancouver style reference format to be used for listing the references in the article. Subsequently, alterations to be made in Reference 1, 7,8 and 14.

Some formatting required in the participant section in the Abstract to ensure that only the necessary subject headings are highlighted.

Reviewer #2: The manuscript seem to be suitable for publication. It could be interesting for the readers of the journal.

7. PLOS authors have the option to publish the peer review history of their article (what does this mean?). If published, this will include your full peer review and any attached files.

Reviewer #1: No

Reviewer #2: **Yes: **Mohammad Asad

---

## [Author Response · Author response to Decision Letter 1]

11 Jul 2022

Response to Reviewer #1: 

We are glad to receive your further feedback. We have made corresponding changes according to your comments. Specifically, we have updated the Vancouver format that meets the requirements of the "PLOS ONE" format, while updating some real-time statistics and replacing some more appropriate references.

---

## [Decision Letter · Decision Letter 2]

4 Aug 2022

Clinical characteristics of patients with confirmed and asymptomatic SARS-CoV-2 infection in China

PONE-D-21-19790R2

Dear Dr. He,

We’re pleased to inform you that your manuscript has been judged scientifically suitable for publication and will be formally accepted for publication once it meets all outstanding technical requirements.

Kind regards,

Academic Editor

PLOS ONE

Additional Editor Comments (optional):

Reviewers' comments:

Reviewer's Responses to Questions

**Comments to the Author**

1. If the authors have adequately addressed your comments raised in a previous round of review and you feel that this manuscript is now acceptable for publication, you may indicate that here to bypass the “Comments to the Author” section, enter your conflict of interest statement in the “Confidential to Editor” section, and submit your "Accept" recommendation.

Reviewer #1: All comments have been addressed

Reviewer #2: All comments have been addressed

2. Is the manuscript technically sound, and do the data support the conclusions?

Reviewer #1: Yes

Reviewer #2: Partly

3. Has the statistical analysis been performed appropriately and rigorously? 

Reviewer #1: Yes

Reviewer #2: I Don't Know

4. Have the authors made all data underlying the findings in their manuscript fully available?

Reviewer #1: Yes

Reviewer #2: Yes

5. Is the manuscript presented in an intelligible fashion and written in standard English?

Reviewer #1: Yes

Reviewer #2: Yes

6. Review Comments to the Author

Reviewer #1: The article titled, "Clinical characteristics of patients with confirmed and asymptomatic SARS-CoV-2 infection in China" may be considered suitable for publication.

Reviewer #2: (No Response)

7. PLOS authors have the option to publish the peer review history of their article (what does this mean?). If published, this will include your full peer review and any attached files.

Reviewer #1: No

Reviewer #2: **Yes: **Mohammad Asad

---

## [Editor Report · Acceptance letter]

12 Aug 2022

PONE-D-21-19790R2 

Clinical characteristics of patients with confirmed and asymptomatic SARS-CoV-2 infection in China 

Dear Dr. He:

I'm pleased to inform you that your manuscript has been deemed suitable for publication in PLOS ONE. Congratulations! Your manuscript is now with our production department. 

Kind regards, 

on behalf of

Dr. Robert Jeenchen Chen 

Academic Editor

PLOS ONE